# High Serum Levels of Toxin A Correlate with Disease Severity in Patients with *Clostridioides difficile* Infection

**DOI:** 10.3390/antibiotics10091093

**Published:** 2021-09-10

**Authors:** Guido Granata, Davide Mariotti, Paolo Ascenzi, Nicola Petrosillo, Alessandra di Masi

**Affiliations:** 1Clinical and Research Department for Infectious Diseases, National Institute for Infectious Diseases L. Spallanzani, IRCCS, 00149 Rome, Italy; guido.granata@inmi.it (G.G.); davide.mariotti16@gmail.com (D.M.); nicola.petrosillo@inmi.it (N.P.); 2Department of Science, Section of Biomedical Sciences and Technologies, Roma Tre University, 00146 Rome, Italy; paolo.ascenzi@uniroma3.it; 3Accademia Nazionale dei Lincei, 00165 Rome, Italy

**Keywords:** *Clostridium difficile*, *Clostridioides difficile*, TcdA, TcdB, toxin, toxemia, plasma, method, quantification, CDI severity

## Abstract

*Cloistridioides difficile* (CD) represents a major public healthcare-associated infection causing significant morbidity and mortality. The pathogenic effects of CD are mainly caused by the release of two exotoxins into the intestine: toxin A (TcdA) and toxin B (TcdB). CD infection (CDI) can also cause toxemia, explaining the systemic complications of life-threatening cases. Currently, there is a lack of sensitive assays to detect exotoxins circulating in the blood. Here, we report a new semi-quantitative diagnostic method to measure CD toxins serum levels. The dot-blot assay was modified to separately detect TcdA and TcdB in human serum with a limit of detection at the pg/mL levels. TcdA and TcdB concentrations in the plasma of 35 CDI patients were measured at the time of CDI diagnosis and at the fourth and tenth day after CDI diagnosis and initiation of anti-CDI treatment. TcdA and TcdB levels were compared to those determined in nine healthy blood donors. Toxemia was detected in the plasma of 33 out of the 35 CDI cases. We also assessed the relationship between TcdA serum levels and CDI severity, reporting that at the time of CDI diagnosis the proportion of severe CDI cases with a TcdA serum level > 60 pg/µL was higher than in mild CDI cases (29.4% versus 66.6%, *p* = 0.04). In conclusion, data reported here demonstrate for the first time that toxemia is much more frequent than expected in CDI patients, and specifically that high serum levels of TcdA correlate with disease severity in patients with CDI.

## 1. Introduction

The Gram-positive anaerobic bacterium *Clostridioides difficile* (CD) is a leading cause of nosocomial diarrhea worldwide, accounting for 15% of all such infections and resulting in significant morbidity, mortality, and prolonged hospital stay [1,2,3]. The clinical manifestations of CD infection (CDI) range from mild diarrhea to pseudomembranous colitis and toxic megacolon that is relatively uncommon but is associated with high morbidity and mortality [4]. A relevant issue during CDI is its high rate of recurrences (rCDI). Clinical studies show wide-ranging rCDI rates after the primary CDI, between 12% and 40% [5,6]. rCDI is associated with a higher risk of death and higher hospitalization costs [1].

The pathogenic effects of CD are mainly caused by the production of two exotoxins into the intestine: toxin A (TcdA) and toxin B (TcdB) [2,7]. TcdA and TcdB monoglucosylate and inactivate the Rho GTPases of host cells, causing several cytopathic effects, and ultimately colonocyte death and loss of intestinal barrier function [8,9]. Moreover, CD toxins induce an inflammatory response through the recruitment of neutrophils and mastocytes and the consequent release of cytokines [10].

CD toxins can reach the bloodstream of CDI patients, thus causing toxemia, which may explain the systemic complications of life-threatening cases [9,11]. To date, only three toxemia-positive cases (two adults and one child) have been reported in humans [12,13]. As the currently available tests for the detection of free toxins are in development and have low sensitivity [3,14,15], CDI-induced toxemia goes undetected in common clinical practice [9,11,15]. Therefore, the lack of a sensitive detection method to determine CD toxin levels in the blood or other body fluids makes it difficult to investigate the relationship between toxemia and systemic clinical manifestations of CDI [12].

Here, we performed a pilot study in which we report and validate a new diagnostic tool to measure serum levels of CD toxins and to define the relationship between the toxemia level and the degree of clinical severity in CDI patients. Setting up this sensitive method allowed us to also highlight that toxemia is much more frequent than expected in CDI patients, and specifically that high serum levels of TcdA correlate with disease severity in patients with CDI.

## 2. Results

### 2.1. Method Set-Up

With the aim of determining the levels of TcdA and TcdB in the serum of CDI patients, several preliminary tests were performed to set up all the parameters necessary to obtain the most sensitive results. In the beginning, a wide range of TcdA and TcdB concentrations were tested (i.e., 2000 pg/µL, 200 pg/µL, 20 pg/µL, 2 pg/µL, 0.2 pg/µL, 0.02 pg/µL, 0.002 pg/µL, 0.0002 pg/µL, and 0.00002 pg/µL (data not shown). Given the different sensitivities of the two primary anti-TcdA and anti-TcdB antibodies used, the scales of the standard curves were adapted to the single toxin, reaching the optimal conditions of 200, 500, 1000, 1500, and 2000 pg/µL for TcdA, and 2, 5, 10, 20, 50, and 100 pg/µL for TcdB (Figure 1). Toxin concentrations below which it was not possible to obtain a quantifiable signal and above which the signal was saturated were excluded.

Besides the definition of the optimal TcdA and TcdB concentrations required to set up the standard curve, we also verified that the resuspension of *C. difficile* toxins in PBS prior to the dot-blot caused the complete loss of the signal. Toxin resuspension in TBS allowed for the restoration of the required sensitivity (Appendix A). Indeed, as Tris-HCl allows lipids to be resuspended [16,17], the results obtained demonstrated that the choice of the buffer is critical to allow the solubilization of the serum lipid fraction (e.g., albumin, lipoproteins, and chylomicrons) present in the samples.

We also demonstrated the absence of antibody cross-reactivity events, supporting the specificity of the assay for the detection of TcdA and TcdB (Appendix A).

### 2.2. Clinical Features of CDI Patients

The demographic and epidemiological data, comorbidities, clinical characteristics, and outcome of the 35 CDI patients enrolled in this study are reported in Appendix A. Among them, 18 (51.4%) patients were female. The mean age of the 35 CDI patients was 60 years, ranging between 19 and 86 years. The mean age-adjusted Charlson co-morbidity index (CCI) at admission was 3.6, ranging between 0 and 8. Among the 35 CDI patients, 30/35 (85.7%) cases had a primary CDI and 5/35 (14.3%) had an rCDI. Regarding CDI severity, 17/35 (48.6%) and 18/35 (51.4%) cases had mild and severe CDI, respectively.

At admission, 25/35 (71.4%) patients reported at least one comorbidity, including cardiovascular disease, chronic obstructive pulmonary disease (COPD), diabetes, and immunodeficiency in 13/35 (37.1%), 9/35 (25.7%), 7/35 (20.0%), and 7/35 (20.0%), respectively. Regarding patient outcomes, 34/35 (97.1%) CDI cases were discharged without complications; one patient (2.8%) died in the hospital. Therefore, 34 patients were followed up for 30 days from the hospital discharge. Of these 34 patients, 28/34 (82.4%) fully recovered at home, presenting no subsequent rCDI. Five patients (14.7%) developed rCDI and were readmitted to the hospital. One other patient (2.9%) was readmitted to the hospital for reasons other than CDI and died during the hospital stay (Appendix A).

### 2.3. TcdA and TcdB Serum Levels and CDI Severity

The definitions of CDI, microbiological evidence of CD, CDI recurrence, mild CDI, severe CDI, and complicated CDI are reported in Appendix A. The mean laboratory findings, total toxemia, and TcdA and TcdB serum levels at CDI diagnosis (T0), as well as at 4 (T4) and 10 days (T10) after CDI diagnosis of the 35 CDI patients included in this study are reported in Table 1.

Serum levels of TcdA and TcdB in the 35 CDI patients serum enrolled in this study, as well as in the 9 healthy donors, were assessed using the standard curve set up for each CD toxin. Notably, neither TcdA nor TcdB was detected in the serum of the 9 healthy donors. Toxemia was detected at least at one time point in 33 out of the 35 CDI patients and was mainly referable to TcdA (Appendix A). The mean total toxemia serum levels ± standard deviation (SD) was 99.24 ± 103.24 pg/µL, 89.74 ± 78.18 pg/µL, and 57.70 ± 72.70 pg/µL at T0, T4, and T10 (Table 1), respectively. TcdA contributed significantly to toxemia, as the mean value of TcdA serum levels were 92.28 ± 96.08 pg/µL, 83.75 ± 74.81 pg/µL, and 49.96 ± 70.96 pg/µL, at T0, T4, and T10, respectively. On the contrary, the mean value of TcdB serum levels was significantly lower compared to those of TcdA (i.e., 6.96 ± 25.71 pg/µL, 5.99 ± 22.2 pg/µL, and 7.74 ± 22.6 pg/µL at T0, T4, and T10, respectively) and no significant variations were detected over time and with respect to CDI severity (Table 1 and Appendix A).

When the 35 CDI cases were grouped according to CDI severity, we found that at CDI diagnosis (i.e., T0), mild and severe CDI cases had a mean TcdA serum level of 64.68 ± 92.22 pg/µL and 116.81 ± 95.19 pg/µL, respectively, and a mean TcdB serum level of 5.95 ± 16.31 pg/µL and 7.86 ± 32.34 pg/µL, respectively (Table 2). At T4, mild and severe CDI cases had a mean TcdA serum level of 51.76 ± 58.48 pg/µL and 115.74 ± 77.14 pg/µL, respectively (*p* = 0.01), and a mean TcdB serum level of 4.21 ± 12.85 pg/µL and 7.76 ± 29.07 pg/µL, respectively. At T10, mild and severe CDI cases had a mean TcdA serum level of 30.80 ± 52.92 pg/µL and 66.73 ± 81.65 pg/µL and a mean TcdB serum level of 4.59 ± 12.74 pg/µL and 10.50 ± 28.79 pg/µL, respectively (Table 2).

The limited number of patients enrolled in this pilot study and the high variability of CD toxins concentration among CDI patients serum may be responsible for the large SD values determined. To reduce this noise, we performed a separate analysis of TcdA plasma concentration over time in mild and severe CDI cases. As reported in Figure 2, TcdA levels significantly decreased at T10 compared to T0 (*p* = 0.0287) and T4 (*p* = 0.0452) in mild but not in severe CDI. This might suggest that the severity of the symptoms is correlated to the persistence of TcdA in patients’ serum.

To assess the relationship between TcdA serum concentration and CDI severity, CDI cases were divided into two groups according to their TcdA serum levels. We set a cut-off, i.e., serum TcdA ≤ 60 pg/µL and serum TcdA > 60 pg/µL (Table 2). At T0 and T4, the proportion of severe CDI cases with a TcdA serum level > 60 pg/µL was higher than among mild CDI cases (at T0: 29.4% versus 66.6%, *p* = 0.04; at T4: 29.4% versus 66.6%; RR: 1.1, 95% CI: 0.4–2.9; *p* = 0.01) (Table 2). The risk factors for TcdA serum levels > 60 pg/µL and for TcdB detectable levels (>4 pg/µL) at the CDI diagnosis are shown in Appendix A, respectively.

## 3. Discussion

CD represents a major public healthcare-associated infection, causing significant morbidity, mortality, and economic burden [18]. Over the past two decades, CDI diagnostic techniques have changed in line with a greater understanding of CDI physiopathology [18]. However, currently, there is relevant under-diagnosis and misdiagnosis of CDI.

Until recently, the phenomenon of the extra-intestinal damage caused by CD toxins has been undervalued [13]. It has been hypothesized that such a low number of toxemia-positive cases is due to the lack of sensitive toxin detection methods and possibly to the presence in human serum of anti-toxin antibodies [13,15]. To date, assays that allow detection of toxemia are (i) the tissue culture cytotoxicity assay, which displays a limit of detection (LOD) of 1–10 ng/mL for TcdA and 10–100 pg/mL for TcdB [12] and (ii) the ultrasensitive immunocytotoxicity assay, characterized by a LOD of 0.1–1.0 pg/mL for TcdA (when used with enhancing antibody A1H3) and 10.0 pg/mL for TcdB [13,19]. Very recently, an ultrasensitive single molecule array able to detect serum concentrations of TcdA and TcdB separately in the range of pg/mL was developed. However, this method was unable to detect TcdA or TcdB in a large panel of serum samples from CDI patients, including the severe forms [20].

Although circulating neutralizing antibodies could reduce the sensitivity of CD toxins detection in human plasma [13,15], we set up a semi-quantitative method that allows to separately detect TcdA and TcdB plasma levels in the range of pg/μL in CDI patients. Results obtained indicated that TcdA contributed significantly to toxemia. Indeed, TcdA serum levels > 60 pg/µL were measured in the plasma of severe CDI cases, and a statistically significant association between TcdA serum levels higher than 60 pg/µL and CDI severity was found, while no association was observed between TcdB toxemia and CDI severity. Remarkably, no detectable TcdB levels were found in the five rCDI patients. As previously reported, specific blockers present in the serum, such as antibodies or serum proteins, may be responsible for reducing the sensitivity of toxins detection in plasma [7,9,13,15]. Therefore, it cannot be excluded that the complete elimination of these blockers from CDI patients’ serum would further increase the sensitivity of our method.

Other plasma blockers such as serum proteins may cause reduced sensitivity of methods that aim at detecting CDI toxins in plasma. Albumin, which is the most abundant protein in human plasma, is able to bind both TcdA and TcdB with high affinity [21,22]. Indeed, low albumin levels indicate a higher risk of acquiring and developing severe CDI and are associated with recurrent and fatal disease [22,23]. Here, we report higher basal albumin levels in patients with TcdA levels > 60 pg/µL at CDI onset (3.3 versus 3.9 g/dL, *p* = 0.02). This indicates that the semi-quantitative method reported here allows for overcoming the limitations due to albumin. This was possible thanks to the use of a resuspension buffer containing Tris-HCl, which solubilizes lipophilic molecules (e.g., albumin, lipoproteins, and chylomicrons) typically present in serum [16,17]. So, while CD toxins and CDI sera suspended in PBS were not detectable with our semi-quantitative dot-blot method, toxins resuspension in the TBS buffer allowed us to overcome this limitation. Notably, both immunoglobulins and albumin are part of humoral immunity, which plays a crucial role in protecting individuals from severe and/or rCDI [9,24,25,26,27].

To date, the role of TcdA in the pathogenesis of CDI is controversial. One paper suggests that CD mutants lacking TcdB, but expressing TcdA, do not cause colitis [19], whereas two other papers highlighted a key role played by TcdA in the disease [7,28]. These different findings possibly reflect differences in CD strains and in the experimental models of the studies [29]. A possible explanation for our findings is that TcdB tends to bind and concentrate more efficiently in the human gut cells than TcdA, leading to relatively higher TcdB intracellular concentrations and higher TcdA concentrations in the gut lumen. This agrees with histopathological analysis of cecal and colonic tissues collected from infected mice showing that TcdB is responsible for the majority of intestinal damage arising during infection, with TcdA causing more superficial and localized damage [24]. Therefore, the two CD toxins may exert different activities and subsequent pathological effects. Moreover, TcdB has been reported as a more potent enterotoxin than TcdA in human intestinal explants [30]. Therefore, our data highlighting the presence of TcdA in CDI patients serum may be an epiphenomenon of the increased endothelial gut permeability in severe CDI patients. Notably, although higher levels of TcdA than TcdB were detected in the sera of CDI patients with a severe form of CDI, these do not specifically indicate that TcdA is the major toxin in CDI.

The CD binary toxin (CDT) is considered an important additional toxin that plays a role in CDI pathogenesis, as strains producing only CDT have been isolated from patients with colitis [31]. However, only a few strains of CD synthesize CDT in the absence of TcdA and TcdB, and the role of CDT in CDI is still unclear [9,30,31,32,33]. Although we did not quantify CDT plasma levels in the patients enrolled in the present study, we assessed via PCR the presence of the binary toxin encoding gene (i.e., *tcdC*) in the CD strains responsible for CDI. No significant association was found between the presence of the *tcdC* gene and CDI severity (Appendix A).

Among the limitations of this study, the relatively low number of patients enrolled should be mentioned. However, it should be also noted that this was intended as a pilot study providing the means to evaluate the technical aspects of the new sensitive method we have set up while serving as a platform to generate preliminary data and foster investigator development. The possibility to determine CD toxin levels in serum will allow for better evaluation of the patient’s prognosis and to address the clinical efforts towards the management of CDI. For the future, a larger study including a higher number of CDI patients with different levels of disease severity will be necessary to support our data which indicate that the presence of TcdA in the blood of CDI patients may be a more sensitive predictor of the disease severity than TcdB.

## 4. Materials and Methods

This study was performed at the Infection Disease Unit of the National Institute for Infectious Diseases “L. Spallanzani” (Rome, Italy) between January 2019 and August 2020. The study was approved by the Ethic Committee of the hospital (National Institute for Infectious Diseases “L. Spallanzani”, IRCCS, Rome; Ethics Committee registry number: 22/2018). Informed consent was obtained from each enrolled patient.

### 4.1. Study Design

#### 4.1.1. Healthy Blood Donors Enrolment

Nine adult (>18 years old) healthy individuals were voluntarily included in the study as a control group. For each healthy control, a single blood collection and determination of CD serum toxins was performed. To exclude CD colonization, all the controls included in the study underwent CD screening using an enzyme immunoassay test on the same day as the blood collection and CD toxins’ serum level determination. Rectal swabs were obtained when the healthy donors were not able to produce stool samples.

#### 4.1.2. CDI Patient Enrolment

Between January 2019 and August 2020, all the adult patients (age ≥ 18 years) admitted to our Infection Disease Unit with a diagnosis of CDI were prospectively enrolled in this study. CDI diagnosis was made in the presence of a clinical picture characterized by diarrhea or ileum or toxic megacolon and microbiological evidence of CDI (Appendix A). Demographic, epidemiological, and clinical data (age and gender, date of hospital admission, patient comorbidities, CDI onset, and clinical characteristics, medications given for CDI, antimicrobial treatments before and after the diagnosis of CDI, laboratory findings, and patient outcome) were collected in clinical record forms (CRF) by trained healthcare personnel. All the CDI cases were followed up to thirty days after the end of the anti-CDI treatment for the CDI episode to assess for new onset of diarrhea, recurrence of CDI, and mortality. In the case of hospital discharge before the end of follow-up, patients were contacted by phone call.

In order to assess the relationship between CD toxin serum level and the severity and recurrence rate of CDI, for each CDI case, three determinations of serum toxins were performed: (i) T0, which corresponds to the time of CDI diagnosis, before the initiation of the anti-CDI treatment; (ii) T4, which corresponds to the fourth day after CDI diagnosis and initiation of the anti-CDI treatment; and (iii) T10, which corresponds to the tenth day after CDI diagnosis and initiation of the anti-CDI treatment.

### 4.2. Experimental Methodology

#### 4.2.1. Preparation of Serum Samples from Healthy Controls and CDI Patients

Blood samples were collected from healthy controls and CDI patients. CDI patients were diagnosed by enzyme immunoassay. The isolation of the plasma component from serum was performed by centrifuging samples at 700× *g* for 5 min. Samples were immediately stored at −20 °C until dot-blot analysis.

#### 4.2.2. Reagents and Antibodies

All the reagents were purchased from Merck KGaA (Darmstadt, Germany). Recombinant TcdA (BML-G140) and TcdB (BML-G150) were obtained from Enzo Life Sciences (Farmingdale, NY, USA). Toxins were reconstituted in 250 µL of distilled H_2_O to reach a final concentration of 200 ng/µL.

#### 4.2.3. Dot-Blot Analysis

##### Standard Reference

The sensitivity of the method has been assessed using toxemia-negative spiked commercial human serum (Merck KGaA, Darmstadt, Germany). Serial dilutions of TcdA and TcdB were obtained by diluting the 200 ng/mL working solution of each toxin in a mixture composed of 1 µL commercial human serum and 49 µL TBS (100 mM NaCl, 10 mM Tris-HCl pH 7.5, 1 mM EDTA). The following final concentrations, determined specifically for each toxin depending on the sensitivity of the primary anti-TcdA and anti-TcdB antibody, were used to set up the standard curve: 200, 500, 1000, 1500, and 2000 pg/µL for TcdA, and 2, 5, 10, 20, 50, and 100 pg/µL for TcdB. Fifty microliters of each toxin dilution were spotted in triplicate onto a polyvinildiene difluoride (PVDF) membrane. The membrane was then placed into the BioDotTM Microfiltration apparatus (Bio-Rad, Hercules, CA, USA) and subjected to moderate suction for 5 s by vacuum pumping (HydroTechTM Vacuum Pump; Bio-Rad, Hercules, CA, USA). Following aspiration, the membrane was blocked in 5% non-fat dry milk dissolved in TBS/0.1% Tween-20 (TTBS) (*w*/*v*) for 1 h at room temperature (RT). The membrane was hybridized overnight at 4 °C with the anti-*Clostridioides difficile* TcdA (MCA2597; Bio-Rad, Hercules, CA, USA) or the anti-*Clostridioides difficile* TcdB (MCA4737; Bio-Rad, Hercules, CA, USA) primary antibodies diluted 1:20 in 5% non-fat dry milk dissolved in TTBS (*w*/*v*). After membrane washing, filters were incubated for 1 h at RT with the horseradish peroxidase-conjugated goat anti-mouse secondary antibody (#1706516; Bio-Rad, Hercules, CA, USA) diluted 1:1000 in PBS/0.01% Tween-20 (*v*/*v*). After membrane washing, dot signals were visualized using the Clarity™ Western ECL substrate (Bio-Rad, Hercules, CA, USA). Images were acquired using the ChemiDoc™ Imaging system (Bio-Rad, Hercules, CA, USA). Toxin levels were quantified using the Image Lab software (version 2.1.0.35.deb, Bio-Rad, Hercules, CA, USA).

##### Determination of Toxins Concentration in Serum Samples Derived from CDI Patients and Healthy Controls

One microliter of human sera derived from the 9 healthy donors and the 35 CDI patients was diluted in 49 µL TBS (dilution 1:50). The 50 µL solution was spotted in triplicate onto a PVDF membrane, then placed into the BioDotTM Microfiltration apparatus. The next experimental procedure was the same as reported in Section Standard Reference.

### 4.3. Data Analysis

Toxin concentrations are reported as the mean value of triplicate dots ± SD. Quantitative variables were tested for normal distribution and compared by means of paired Student’s *t*-test. Comparisons between CD toxin concentrations in mild and severe CDI cases were performed by One way-ANOVA test. Qualitative differences between groups were assessed by use of Fisher’s exact test. The precision of the risk ratio was determined by calculating a 95% confidence interval. The relationship between categorical variables and CDI severity was assessed by Pearson Chi-Square analysis. The risk ratio and 95% confidence interval were calculated. Statistical analysis was performed using the software programs GraphPad InStat 3.1 Software Inc. (San Diego, CA, USA) and IBM SPSS Statistics for Windows version 24.0 (IBM Corp. Released 2016; Armonk, NY, USA). Results were considered significant when *p* values were ≤0.05.

## 5. Conclusions

In conclusion, setting up this sensitive method allowed us to highlight that toxemia is much more frequent than expected in CDI patients, and more specifically, that high serum levels of TcdA correlate with disease severity in patients with CDI.

## Figures and Tables

**Figure 1 antibiotics-10-01093-f001:**
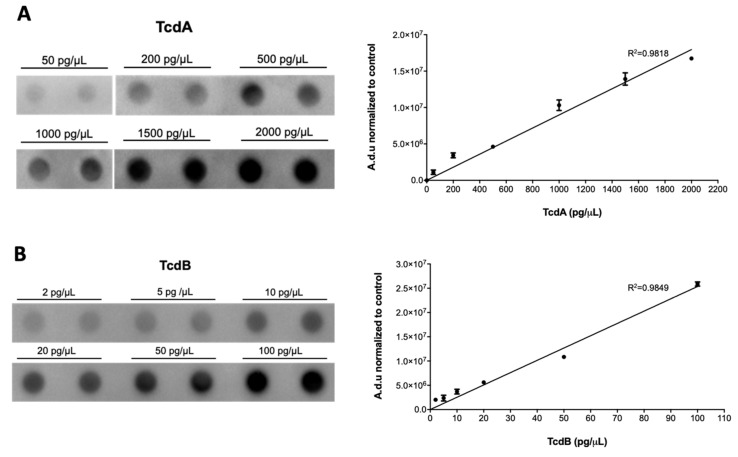
Representative images and the standard curve of (**A**) TcdA and (**B**) TcdB dot plots necessary for determining the concentration of the toxins in the blood of CDI patients. Toxin concentrations are reported as the mean value of triplicate dots ± standard deviation (SD) normalized to control (i.e., 0 pg/mL); a.d.u., arbitrary densitometric unit.

**Figure 2 antibiotics-10-01093-f002:**
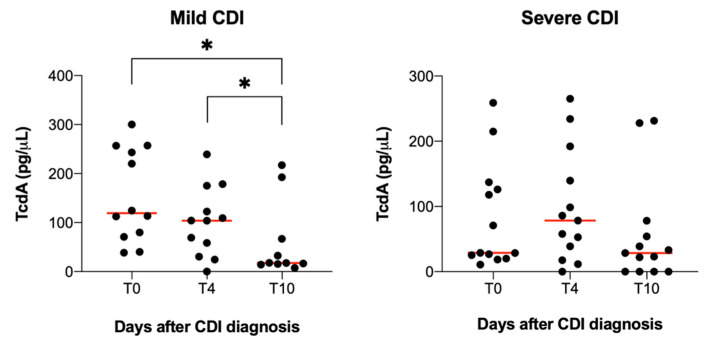
Changes of TcdA plasma levels in mild and severe CDI cases. TcdA levels significantly decreased at 10 days (T10) after CDI diagnosis in mild cases compared to levels measured at T0 and T4. Patients whose TcdA levels at T0 were 0 pg/μL were not included in these graphs. One way-ANOVA test, * *p* < 0.05.

**Table 1 antibiotics-10-01093-t001:** Mean laboratory findings and mean toxemia at CDI diagnosis (T0), and at 4 (T4) and 10 days (T10) after CDI diagnosis of the 35 CDI patients included in the study. SD: standard deviation.

Laboratory Findings	T0	T4	T10
Total white blood cells peripheral count (10^3^ cells/µL ± SD)	11.04 ± 5.99	7.03 ± 2.59	6.33 ± 2.64
Neutrophils peripheral count (10^3^ cells/µL ± SD)	8.18 ± 5.45	4.41 ± 2.25	4.11 ± 2.46
Blood creatinine value (mg/dL ± SD)	1.0 ± 0.6	0.8 ± 0.3	0.8 ± 0.2
Blood albumin value (g/dL ± SD)	3.3 ± 0.5	3.4 ± 0.6	3.6 ± 0.7
Total toxemia (TcdA + TcdB, pg/µL ± SD)	99.24 ± 103.24	89.74 ± 78.18	57.70 ± 72.70
TcdA (pg/µL ± SD)	92.28 ± 96.08	83.75 ± 74.81	49.96 ± 70.96
TcdB (pg/µL ± SD)	6.96 ± 25.71	5.99 ± 22.2	7.74 ± 22.6

**Table 2 antibiotics-10-01093-t002:** Clinical features, comorbidities, laboratory findings, and outcome of the 18 cases with mild CDI and the 17 cases with severe CDI before diagnosis, at T0, T4, and T10. RR: Risk ratio. CI: Confidence interval. SD: Standard deviation. CCI: Charlson Co-morbidity Index.

	Mild CDI (*N* = 17)	Severe CDI (*N* = 18)	RR (95% CI)	Fisher’s Test *
Female gender	11 (64.7%)	7 (38.8%)	0.5 (0.2–1.2)	*p* = 0.1
Mean age (years)	55.7	63.5	-	*p* = 0.2
Mean age-adjusted CCI at admission ± SD	2.4 ± 2.3	4.7 ± 2.9	-	*p* = 0.01
**Comorbidities**				
No comorbidities	7 (41.1%)	3 (16.6%)	1.7 (0.9–3.2)	*p* = 0.1
Cardiovascular disease	4 (23.5%)	9 (50.0%)	1.9 (0.7–4.6)	*p* = 0.1
Heart failure	1 (5.9%)	3 (16.6%)	2.0 (0.3–11.6)	*p* = 0.3
Diabetes	2 (11.7%)	5 (27.7%)	1.8 (0.5–6.3)	*p* = 0.4
Renal failure	0 (0%)	3 (16.6%)	-	*p* = 0.2
Inflammatory bowel disease	1 (5.8%)	1 (5.5%)	0.9 (0.2–4.0)	*p* = 1
Chronic liver failure	1 (5.8%)	1 (5.5%)	0.9 (0.2–4.0)	*p* = 1
Neurological disease	0 (0%)	4 (22.2%)	-	*p* = 0.1
Vasculitis	0 (0%)	2 (11.1%)	-	*p* = 0.4
COPD	6 (35.3%)	3 (16.6%)	0.6 (0.3–1.2)	*p* = 0.2
Solid cancer	1 (5.8%)	2 (11.1%)	1.5 (0.2–7)	*p* = 1
Blood cancer	1 (5.8%)	2 (11.1%)	1.5 (0.2–7.7)	*p* = 1
Transplant, immunodeficiency, immunosuppression	1 (5.8%)	6 (33.3%)	4 (0.6–25.2)	*p* = 0.08
Other bacteria infections at admission	6 (35.3%)	7 (38.8%)	1.0 (0.5–2.2)	*p* = 1
**Laboratory findings before CDI diagnosis**				
Basal Albumin (g/dL ± SD)	3.7 ± 0.6	3.5 ± 0.6	-	*p* = 0.5
Basal Creatinine (mg/dL ± SD)	0.70 ± 0.1	0.8 ± 0.3	-	*p* = 0.2
**Laboratory findings at CDI diagnosis (T0)**				
White blood cell peripheral count (10^3^ cells/µL ± SD)	8.81 ± 3.52	13.15 ± 7.10	-	*p* = 0.02
Neutrophils peripheral count (10^3^ cells/µL ± SD)	5.70 ± 3.14	10.52 ± 6.18	-	*p* = 0.007
Creatinine (mg/dL ± SD)	0.8 ± 0.3	1.2 ± 0.7	-	*p* = 0.02
Albumin (g/dL ± SD)	3.5 ± 0.5	3.2 ± 0.5	-	*p* = 0.2
TcdA (pg/µL ± SD)	64.68 ± 92.22	116.81 ± 95.19	-	*p* = 0.1
TcdB (pg/µL ± SD)	5.95 ± 16.31	7.86 ± 32.34	-	*p* = 0.8
TcdA + TcdB (pg/µL ± SD)	70.63 ± 71.65	124.67 ± 89.23	-	*p* = 0.1
TcdA > 60 pg/µL	5 (29.4%)	12 (66.6%)	2.0 (1.0–4.2)	*p* = 0.04
**Laboratory findings at T4**				
White blood cell peripheral count (10^3^ cells/µL ± SD)	6.95 ± 2.38	7.09 ± 2.84	-	*p* = 0.8
Neutrophils peripheral count (10^3^ cells/µL ± SD)	4.25 ± 2.18	4.56 ± 2.37	-	*p* = 0.6
Creatinine (mg/dL ± SD)	0.7 ± 0.2	0.9 ± 0.3	-	*p* = 0.1
Albumin (g/dL ± SD)	3.6 ± 0.5	3.2 ± 0.6	-	*p* = 0.09
TcdA (pg/µL ± SD)	51.76 ± 58.48	115.74 ± 77.14	-	*p* = 0.01
TcdB (pg/µL ± SD)	4.21 ± 12.85	7.76 ± 29.07	-	*p* = 0.7
TcdA + TcdB (pg/µL ± SD)	55.98 ± 48.17	123.51 ± 79.36	-	*p* = 0.01
TcdA > 60 pg/µL	5 (29.4%)	12 (66.6%)	2.4 (1.08–5.3)	*p* = 0.03
**Laboratory findings at T10**				
White blood cell peripheral count (10^3^ cells/µL ± SD)	6.14 ± 1.81	6.49 ± 3.36	-	*p* = 0.7
Neutrophils peripheral count (10^3^ cells/µL ± SD)	4.06 ± 1.66	4.15 ± 3.05	-	*p* = 0.9
Creatinine (mg/dL ± SD)	0.7 ± 0.2	0.8 ± 0.2	-	*p* = 0.2
Albumin (g/dL ± SD)	3.7 ± 0.7	3.5 ± 0.7	-	*p* = 0.4
TcdA (pg/µL ± SD)	30.80 ± 52.92	66.73 ± 81.65	-	*p* = 0.1
TcdB (pg/µL ± SD)	4.59 ± 12.74	10.50 ± 28.79	-	*p* = 0.4
TcdA + TcdB (pg/µL ± SD)	35.39 ± 40.06	77.23 ± 66.65	-	*p* = 0.1
TcdA > 60 pg/µL	3 (17.6%)	4 (22.2%)	1.1 (0.4–2.9)	*p* = 1
Patients outcome				
Deceased	0 (0%)	1 (5.5%)	0.5 (0.3–0.7)	*p* = 1
rCDI	3 (17.6%)	2 (11.1%)	1.5 (0.2–10.3)	*p* = 1

* Paired *t*-test for non-categorical variables.

## Data Availability

The data presented in this study are available on request from the corresponding author. The data are not publicly available due to privacy and ethical restrictions.

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
