# Peer review of "High Serum Levels of Toxin A Correlate with Disease Severity in Patients with Clostridioides difficile Infection"

_antibiotics, 2021, doi:10.3390/antibiotics10091093_

Round 1
Reviewer 1 Report
This manuscript reports an improved method of C. difficile toxins titration in patient's sera with CD infection. The dot-blot method developed by the authors is specific of each CD toxin, TcdA and TcdB, and allows a detection in patient's sera in the pg/ul range. High levels (>60 pg/ul) of TcdA in patient's sera were associated with severe cases of CDI.
The limitation of this study is the low number of patients.
Higher levels of TcdA than TcdB were detected in the patient's sera with severe form of CDI. However, this does not preclude different toxin activities and subsequent pathological effects. Indeed, TcdB was reported to be much more cytotoxic than TcdA as tested in cultured cells. Thus, it is not clear whether larger amounts of TcdA in sera indicate that TcdA is the major toxin in CDI.
The binary toxin CDT has not been investigated. The epidemic strains responsible for severe cases produce CDT which is considered as an important additional toxin in the severe cases of CDI. Bacteriological investigation including toxin gene distribution in the C. difficile strains involved in the patients of this study would be shown.
Author Response
Reviewer #1
This manuscript reports an improved method of C. difficile toxins titration in patient's sera with CD infection. The dot-blot method developed by the authors is specific of each CD toxin, TcdA and TcdB, and allows a detection in patient's sera in the pg/ul range. High levels (>60 pg/ul) of TcdA in patient's sera were associated with severe cases of CDI.
R1.1 The limitation of this study is the low number of patients.
A1.1. We fully agree with the Reviewer that the main limitation of this study is the low number of patients. However, this was intended as a pilot study providing the means to evaluate the technical aspects of the new sensitive method we have set up while serving as a platform to generate preliminary data and foster investigator development. Indeed, this study aimed at evaluating the validity of our assay in quantifying serum levels of C. difficile toxins as well as to define the relationship between the toxemia level and the degree of clinical severity in CDI patients. As N=30 is recognized as a reasonable minimum sample size for bootstrapped confidence intervals, results obtained by this pilot study opens the door to a further study involving a larger study.
The further aspect that should be noted is that we faced difficulties in enlarging the study cohort during the ongoing COVID-19 pandemic, as our hospital has been now entirely dedicated to the management of COVID-19 patients.
In the revised version of the manuscript, we specified that this was intended as a pilot study in the Introduction (page 2, line 60) and we added specific statements to the Discussion sections (page 8, lines 253-257 and lines 259-262).
R1.2. Higher levels of TcdA than TcdB were detected in the patient's sera with severe form of CDI. However, this does not preclude different toxin activities and subsequent pathological effects. Indeed, TcdB was reported to be much more cytotoxic than TcdA as tested in cultured cells. Thus, it is not clear whether larger amounts of TcdA in sera indicate that TcdA is the major toxin in CDI.
A1.2. We added to the Discussion of the revised manuscript two comments related to this point (page 8, lines 239-240 and lines 242-244).
R1.3. The binary toxin CDT has not been investigated. The epidemic strains responsible for severe cases produce CDT which is considered as an important additional toxin in the severe cases of CDI. Bacteriological investigation including toxin gene distribution in the C. difficile strains involved in the patients of this study would be shown.
A1.3. We agree with the Reviewer’s observation that binary toxin CDT is an important additional virulence factor during CDI. We were able to retrieve this data and we have now added the information related to the presence of the binary toxin tcdC gene by polymerase chain reaction to the Supplementary Table S3, for all the CDI patients included in the study. In addition, we added specific comments to this point in the Discussion of the revised manuscript (page 8, lines 245-252).
Reviewer 2 Report
Granata etal has presented a study highlighting the significance of detection of Clostridioides difficile toxin A and its relation to severe CDI. Although the ultralow levels of toxin A in circulating blood correlated well with disease severity, the small sample size (n=35) warrants confirmation of these findings in a relatively larger population. Confirmation of the toxin A levels using another ultrasensitive method performed in parallel would have substantiated the findings even with this small sample size. The role of circulating neutralizing antibodies in the detection of toxin has been highlighted by other researchers. The role of these antibodies in test performance could be mentioned as a limitation of the study.
Author Response
Reviewer #2
R2.1. Granata et al has presented a study highlighting the significance of detection of Clostridioides difficile toxin A and its relation to severe CDI. Although the ultralow levels of toxin A in circulating blood correlated well with disease severity, the small sample size (n=35) warrants confirmation of these findings in a relatively larger population. Confirmation of the toxin A levels using another ultrasensitive method performed in parallel would have substantiated the findings even with this small sample size.
A2.1 We fully agree with the Reviewer that the main limitation of this study is the low number of patients. However, this was intended as a pilot study providing the means to evaluate the technical aspects of the new sensitive method we have set up while serving as a platform to generate preliminary data and foster investigator development. Indeed, this study aimed at evaluating the validity of our assay in quantifying serum levels of C. difficile toxins as well as to define the relationship between the toxemia level and the degree of clinical severity in CDI patients. As N=30 is recognized as a reasonable minimum sample size for bootstrapped confidence intervals, results obtained by this pilot study opens the door to a further study involving a larger study. The further aspect that should be noted is that we faced difficulties in enlarging the study cohort during the ongoing COVID-19 pandemic, as our hospital has been now entirely dedicated to the management of COVID-19 patients.
In the revised version of the manuscript, we specified that this was intended as a pilot study in the Introduction (page 2, line 60) and we added specific statements to the Discussion sections (page 8, lines 253-257 and lines 259-262).
R2.2. The role of circulating neutralizing antibodies in the detection of toxin has been highlighted by other researchers. The role of these antibodies in test performance could be mentioned as a limitation of the study.
A2.2. Circulating neutralizing antibodies could reduce the sensitivity of our method, although we were still able to quantify TcdA and TcdB plasma levels in the range of pg/mL in CDI patients. However, it cannot be excluded that the complete elimination of serum proteins and circulating antibodies from CDI patients serum would further increase the sensitivity of our method. According to the Reviewer’s suggestion, this possible limitation has been highlighted in the Discussion (page 8, lines 211-214).
Reviewer 3 Report
Thank you for submitting your manuscript in Antibiotics. The manuscript is well written and summarized. However, the study is carried between Jan 2019 to August 2020, it would have been nice if the authors could have chosen a high subject number with different age groups.
I would suggest elaborating the introduction part; however, restrict your information to 2-3 paras.
Author Response
Reviewer #3
R3.1 Thank you for submitting your manuscript in Antibiotics. The manuscript is well written and summarized. However, the study is carried between Jan 2019 to August 2020, it would have been nice if the authors could have chosen a high subject number with different age groups.
A3.1. We fully agree with the Reviewer that the main limitation of this study is the low number of patients. However, this was intended as a pilot study providing the means to evaluate the technical aspects of the new sensitive method we have set up while serving as a platform to generate preliminary data and foster investigator development. Indeed, this study aimed at evaluating the validity of our assay in quantifying serum levels of C. difficile toxins as well as to define the relationship between the toxemia level and the degree of clinical severity in CDI patients. As N=30 is recognized as a reasonable minimum sample size for bootstrapped confidence intervals, results obtained by this pilot study opens the door to a further study involving a larger study. The further aspect that should be noted is that we faced difficulties in enlarging the study cohort during the ongoing COVID-19 pandemic, as our hospital has been now entirely dedicated to the management of COVID-19 patients.
In the revised version of the manuscript, we specified that this was intended as a pilot study in the Introduction (page 2, line 60) and we added specific statements to the Discussion sections (page 8, lines 253-257 and lines 259-262).
R3.2. I would suggest elaborating the introduction part; however, restrict your information to 2-3 paras.
A3.2. According to the Reviewer’s suggestion, the Introduction has been revised with the aim to maintain it concise and clear. Indeed, we preferred to elaborate and revise the Discussion section also in the light of the comments raised by the Reviewers.
Reviewer 4 Report
The manuscript entitled "High toxin A serum levels correlate with disease severity in patients with Clostridioides difficile infection" by Granata et al reported a new semi-quantitative diagnostic method to measure CD toxins serum level. The authors validated their method in patients with CDI. Moreover, the authors found correlation between TcdA serum level and severity of CDI. Overall, this is an interesting paper, and it falls in the scope of Antibiotics. The manuscript is acceptable after addressing the reviewers' following minor concerns:
1. It's more common to use dots rather than comma in decimal numbers. Moreover, the authors were using comma and dots interchangably in the manuscript, which should be consistent. E.g. line 138.
2. In methods, the authors should provide company state/country information for reagents and equipment, not just company names.
3. Line 273, H2O -> 2 should be subscript
4. In table 1, the value of total toxemia, TcdA and TcdB in T0, T4 and T10 have large SD, even several times larger than the mean. This incidates heavy noise and poor reproducibility. I would recommend the authors perform more repeats and get more reliable data.
5. Lack of statistical analysis in table 1 to show significant difference between TcdA and TcdB
Author Response
Reviewer #4
The manuscript entitled "High toxin A serum levels correlate with disease severity in patients with Clostridioides difficile infection" by Granata et al reported a new semi-quantitative diagnostic method to measure CD toxins serum level. The authors validated their method in patients with CDI. Moreover, the authors found correlation between TcdA serum level and severity of CDI. Overall, this is an interesting paper, and it falls in the scope of Antibiotics. The manuscript is acceptable after addressing the reviewers' following minor concerns.
R4.1. It’s more common to use dots rather than comma in decimal numbers. Moreover, the authors were using comma and dots interchangably in the manuscript, which should be consistent. E.g. line 138.
A4.1. We apologize for this oversight. In the revised version of the manuscript, commas have been replaced by dots in decimal numbers throughout the whole text.
R4.2. In methods, the authors should provide company state/country information for reagents and equipment, not just company names.
A4.2. In the original version of the manuscript, companies’ city and state/country were indicated the first time they were mentioned in the text. According to the Reviewer request, in the revised version of the manuscript the company’s city and state/country have been indicated every time they have been mentioned.
R4.3. Line 273, H2O -> 2 should be subscript.
A4.3. This typing error that has been corrected in the revised manuscript.
R4.4. In table 1, the value of total toxemia, TcdA and TcdB in T0, T4 and T10 have large SD, even several times larger than the mean. This indicates heavy noise and poor reproducibility. I would recommend the authors perform more repeats and get more reliable data.
A4.4. We agree with the Reviewer that SD were quite large. However, this was intended as a pilot study providing the means to evaluate the technical aspects of the new sensitive method we have set up while serving as a platform to generate preliminary data and foster investigator development. Indeed, this study aimed at evaluating the validity of our assay in quantifying serum levels of C. difficile toxins as well as to define the relationship between the toxemia level and the degree of clinical severity in CDI patients. As N=30 is recognized as a reasonable minimum sample size for bootstrapped confidence intervals, results obtained by this pilot study opens the door to a further study involving a larger study. The limited number of patients enrolled in this pilot study and the high variability of CD toxins plasma concentration among CDI patients can be responsible for the large SD values determined. To reduce this noise, we performed a separate analysis of TcdA plasma concentration over time in mild and severe CDI cases. As reported in Figure 2, TcdA levels significantly decreased at T10 compared to T0 (P = 0.0287) and T4 (P = 0.0452) in mild but not in severe CDI. This might suggest that the severity of the symptoms is correlated to the persistence of TcdA in patients serum. This analysis has been reported at page 4, lines 149-155).
R4.5. Lack of statistical analysis in table 1 to show significant difference between TcdA and TcdB.
A4.5. We thank the Reviewer for highlighting this point that prompted us to perform a statistical analysis regarding variations of TcdA concentrations over time, as reported in the previous point.
Round 2
Reviewer 1 Report
The revised manuscript was improved and seems suitable for publication.
Please, define n/a in Table S3
Reviewer 4 Report
All my concerns have been addressed.